# The Mitochondrion: A Promising Target for Kidney Disease

**DOI:** 10.3390/pharmaceutics15020570

**Published:** 2023-02-08

**Authors:** Cem Tanriover, Sidar Copur, Duygu Ucku, Ahmet B. Cakir, Nuri B. Hasbal, Maria Jose Soler, Mehmet Kanbay

**Affiliations:** 1Department of Medicine, Koc University School of Medicine, 34010 Istanbul, Turkey; 2Department of Medicine, Division of Nephrology, Koc University School of Medicine, 34010 Istanbul, Turkey; 3Nephrology and Kidney Transplant Research Group, Vall d’Hebron Research Institute (VHIR), 08035 Barcelona, Spain

**Keywords:** mitochondrial dysfunction, acute kidney injury, chronic kidney disease

## Abstract

Mitochondrial dysfunction is important in the pathogenesis of various kidney diseases and the mitochondria potentially serve as therapeutic targets necessitating further investigation. Alterations in mitochondrial biogenesis, imbalance between fusion and fission processes leading to mitochondrial fragmentation, oxidative stress, release of cytochrome c and mitochondrial DNA resulting in apoptosis, mitophagy, and defects in energy metabolism are the key pathophysiological mechanisms underlying the role of mitochondrial dysfunction in kidney diseases. Currently, various strategies target the mitochondria to improve kidney function and kidney treatment. The agents used in these strategies can be classified as biogenesis activators, fission inhibitors, antioxidants, mPTP inhibitors, and agents which enhance mitophagy and cardiolipin-protective drugs. Several glucose-lowering drugs, such as glucagon-like peptide-1 receptor agonists (GLP-1-RA) and sodium glucose co-transporter-2 (SGLT-2) inhibitors are also known to have influences on these mechanisms. In this review, we delineate the role of mitochondrial dysfunction in kidney disease, the current mitochondria-targeting treatment options affecting the kidneys and the future role of mitochondria in kidney pathology.

## 1. Introduction

Acute kidney injury (AKI) is the sudden loss of kidney function with a rise in creatinine and blood urea nitrogen (BUN) which occurs within a few hours or a few days [1]. In addition to the kidneys, AKI can also negatively impact other organ systems including the brain, heart, and lungs, causing significant morbidity [2]. Chronic kidney disease (CKD) is defined by derangements in kidney structure or function lasting a long period of time (i.e., three months or longer) [3]. The etiologies of CKD include diabetes, hypertension, glomerulonephritis, chronic tubulointerstitial nephritis, hereditary or cystic diseases, heart diseases, and stroke [4,5]. AKI and CKD are closely linked in that AKI is an important contributor to CKD development and CKD predisposes patients to AKI [3]. Both conditions represent a global public health problem and thus a better understanding of the pathophysiological mechanisms underlying AKI and CKD is necessary for improving our therapeutic strategies and developing novel agents for these diseases.

The kidney has the second highest level of mitochondria and oxygen use after the heart and is, thus, one of the most energy necessitating organs in the human body [3,6]. Mitochondria are intracellular organelles that are crucial in the generation of adenosine triphosphate (ATP), the regulation of various catabolic and anabolic processes, the maintenance of intracellular calcium, redox homeostasis and production of reactive oxygen species (ROS), thermogenesis, and the regulation of proliferation and intrinsic apoptotic pathways [3,7]. Therefore, the proper maintenance of mitochondria is essential for normal cell function.

Mitochondrial dysfunction has an important role in the pathogenesis of various kidney diseases and the mitochondria potentially serve as therapeutic targets necessitating further investigation. Alterations in mitochondrial biogenesis, imbalance between fusion and fission processes leading to mitochondrial fragmentation, oxidative stress, release of cytochrome c and mitochondrial DNA resulting in apoptosis, mitophagy, and defects in energy metabolism are the primarily areas of research regarding the role of mitochondrial dysfunction in kidney diseases [3,7]. Several agents which target different mitochondrial processes have recently emerged as potential therapeutic approaches in kidney pathology. According to their mechanisms, these compounds can be classified as biogenesis activators, fission inhibitors, antioxidants, mitochondrial permeability transition pore (mPTP) inhibitors, as well as agents which enhance mitophagy and drugs which protect cardiolipin [3,7]. In addition, several glucose lowering drugs such as sodium glucose co-transporter-2 (SGLT-2) inhibitors and glucagon-like peptide-1 receptor agonists (GLP-1-RA) have been shown to act on these mitochondrial processes [8].

In this review, we delineate the role of the mitochondria in kidney disease. We initially explain the pathophysiological mechanisms underlying mitochondrial dysfunction in kidney disease. We then discuss the potential therapeutic approaches which target different mitochondrial processes to preserve and restore mitochondrial function as well as improve treatment of patients with kidney diseases while highlighting the current clinical trials investigating these mitochondria-targeted therapies. In addition, we explain the novel developments in mitochondrial medicine.

## 2. Mitochondria as a Key Regulator of Kidney Disease: The Pathophysiological Basis

Mitochondrial dysfunction has recently been linked to the pathophysiology of various forms of kidney diseases and potentially provides an alternative approach to the treatment of kidney pathology. Alterations in mitochondrial biogenesis, imbalances between fusion and fission processes leading to mitochondrial fragmentation, oxidative stress, release of cytochrome c and mitochondrial DNA resulting in apoptosis, mitophagy, and defects in energy metabolism are the primarily focuses of research regarding the role of mitochondrial dysfunction in kidney diseases. Importantly, structural and functional mitochondrial dysfunction occurs significantly earlier than any detectable renal dysfunction as evidenced by the detection of mitochondrial ultrastructure defects 3 h following the injection of glycerol that persists for up to 144 h in glycerol-induced AKI mice models [9,10]. Additionally, mitochondrial protection through various mechanisms has shown to be protective against AKI if administered before the onset of kidney injury or protective against CKD transition if administered following kidney injury [11,12,13]. Therefore, the appreciation of mitochondrial dynamics is of the utmost importance for the better understanding, prevention, and treatment of kidney diseases (Figure 1).

### 2.1. Alterations of Mitochondrial Biogenesis

Peroxisome proliferator-activated receptor gamma coactivator 1-alpha (PGC-1α), a crucial mediator of mitochondrial metabolism and biogenesis, has been implicated in the pathophysiology of AKI and CKD. PGC-1α is predominantly expressed in the renal cortex and at the corticomedullary junction, areas of the kidney with the highest metabolic demand and cellular respiration [14]. PGC-1α is directly involved in the regulation of multiple transcription factors which take part in mitochondrial biogenesis and metabolism, including peroxisome proliferator-activated receptor-α (PPARα), steroid hormone receptor ERR1, transcriptional repressor protein YY1, nuclear factor erythroid 2-related factor 2 (NRF2), and nuclear respiratory factor 1 (NRF1) [15]. PGC-1α has been implicated in the pathophysiology of AKI, AKI-to-CKD transition, and CKD through its effects on oxidative phosphorylation, fatty acid beta oxidation, and mitochondrial biogenesis [16].

#### 2.1.1. PGC-1α and AKI

Mitochondrial dysfunction is almost universally detected in the setting of AKI, whereas tissue oxygenation does not appear to be altered in such cases [14]. Mitochondrial swelling is a common and early feature of renal ischemia while most forms of AKI, including toxic, inflammatory, and ischemic types have been associated with cortical triglyceride accumulation which may eventually become peroxidated [17,18,19]. The most common features include decreased fusion and increased fragmentation of the mitochondria, increased release of cytochrome C and ROS and exacerbation of apoptosis, impaired production of electron transport chain proteins, impaired depolarization of mitochondrial inner membrane, and decline in mitochondrial nicotinamide adenine dinucleotide (NAD+) levels [20,21,22,23]. Such shifts in mitochondrial dynamics may be prevented or reduced by the activation of sirtuin 1 (SIRT1) or suppression of fission mediator dynamin-related protein 1 (DRP1) as shown by a study conducted on murine models of cisplatin-induced AKI, in which adenosine monophosphate-activated protein kinase (AMPK) agonist 5-aminoimidazole-4-carboxamide-1-β-D-ribofuranoside (AICAR), or the antioxidant agent ALCAR, have restored both SIRT3 expression and renal function [21,24]. Additionally, SIRT3-knockout murine models experienced more severe forms of AKI when given cisplatin [21].

PGC-1α expression has been shown to be reduced in the setting of AKI along with its downstream molecules, as evidenced by studies conducted on septic mice models, demonstrating a direct correlation between the expression levels and the degree of insult which is reversed by the resolution of insult [14]. Moreover, studies have demonstrated that PGC-1α knockout mouse models are more susceptible to AKI from various causes while renal tubule specific overexpression of PGC-1α has been linked to AKI resistance, especially in cases of cisplatin-induced AKI and ischemia-reperfusion injury [25,26,27]. Similarly, upstream activators of PGC-1α including AICAR, which is a small activator molecule of AMPK, and resveratrol, which is an activator of sirtuin, have reduced the severity of AKI [28,29,30]. Levels of such activators of PGC-1α are also reduced in AKI as expected [31,32].

#### 2.1.2. PGC-1α and Glomerular Diseases

The regulation of PGC-1α has widely been investigated in patients and animal models of glomerular diseases, especially diabetic kidney disease. Renal cortex samples from patients with diabetic kidney disease demonstrate reduced expression of PGC-1α which may be attributable to the downregulation of PGC-1α regulators, such as sirtuins and FOXO1 [33,34,35]. RNA sequencing studies revealed that through the expression of a long non-coding RNA, referred to as taurine-upregulated gene 1 (Tug1), an upstream activator of PGC-1α was reduced in the diabetic glomeruli and podocytes while podocyte-specific over-expression of Tug1 resulted in improved histological changes in response to hyperglycemia [36]. Another mediator of PGC-1α in diabetic kidney disease is the PKM2, an enzyme catalyzing the last step of glycolysis, which is downregulated in such patients while podocyte-specific PKM2-knockout animal models showed worse histopathological features [37].

As activation of PGC-1α downstream pathways results in a reduction in proteinuria through increased expression of nephrin and synaptopodin, the potential role of PGC-1α has been investigated in mouse models with nephrotic syndrome [38,39]. Moreover, PGC-1α has been implicated in the pathophysiology of renal fibrosis through various pathways [40,41,42]. However, there is a clear need for future studies for the better understanding of the exact underlying pathophysiology.

### 2.2. Mitochondrial Fusion-Fission Imbalance

Mitochondrial fragmentation via the imbalance between the fusion and fission processes is a key event in the pathophysiology of AKI, AKI-to-CKD transition, and CKD. Mitochondrial fission has a role during cell division and apoptosis while mitochondrial fusion is involved in the optimization of mitochondrial function and reduction in organelle stress [43]. Mitochondrial fission is initiated by the recruitment of DPP-1 to the outer mitochondrial membrane via its receptors, mitochondrial fission factor (MFF), mitochondrial dynamics protein MID49 and/or MID51 at the mitochondria-endoplasmic reticulum junction, followed by the oligomerization of DPP-1 to form a ring-like structure to constrict and facilitate the fission along with dynamin-2 [44,45,46]. On the other hand, mitochondrial fusion involves the fusion of outer membranes facilitated by mitofusin-1 and 2 (MFN-1/2) and fusion of inner membranes mediated by OPA1 [47].

The protein DPP-1, which is involved in the fission of mitochondria, is upregulated while two proteins, namely MFN and OPA1, which are involved in the mitochondrial fusion, are downregulated in kidney diseases [11,48]. Additionally, the oligomerization of Bax and Bak followed by the interaction with the mitochondrial outer membrane results in increased mitochondrial outer membrane permeability and the release of pro-apoptotic factors, such as cytochrome c [49,50,51]. Pharmacological or genetical modifications preventing mitochondrial fragmentation has shown to be reno-protective in animal models of AKI in response to ischemia-reperfusion or nephrotoxic agents [12,52,53,54]. Similarly, mitochondrial fragmentation has reported to be a potential marker for the assessment of disease progression in autosomal dominant polycystic kidney disease in animal models [55].

### 2.3. Mitochondrial DNA as DAMPs

Mitochondrial DNA (mtDNA) released as a result of mitochondrial fragmentation acts as a damage-associated molecular patterns (DAMPs) molecule and leads to the activation of innate and adaptive immune responses and infiltration of tissue via inflammatory cells [56]. mtDNA leads to the activation of cyclic guanosine monophosphate–adenosine monophosphate (GMP–AMP) synthase (cGAS)-stimulator of interferon genes (STING) pathway which causes an unfolded protein response [33,57,58]. This response results in the activation of mPTP leading to apoptosis and the release of interferon gamma, which results in the infiltration of tissues via pro-inflammatory macrophages [56].

Additionally, mtDNA is highly vulnerable to oxidative stress, mainly due to its lack of histone protection and close location to ROS production. Mitophagy and autophagy are predominant pathways of mtDNA degradation, though they are not the only pathways [59]. Oxidized mtDNA has been shown to be damaged in animal models with ischemia-reperfusion injuries shortly following reperfusion, sepsis-induced AKI, and cisplatin-induced AKI, which is evident from the elevated levels of 8-hydroxy-2-deoxy-guanosine as a marker for oxidative DNA damage [60,61,62,63].

### 2.4. Mitophagy

Mitophagy is a selective autophagy process in which accumulated dysfunctional mitochondria are removed. The first signal is the loss of the mitochondrial inner membrane potential [64]. Following mitochondrial damage, the serine/threonine-protein kinase PINK1, which is constitutively being imported into the mitochondrial matrix, is recruited in the outer mitochondrial membrane. Accumulated PINK1 recruits parkin molecules at outer mitochondrial membrane and promotes its E3 ligase activity through phosphorylation, resulting in the building of poly-ubiquitin chains on the outer membrane acting as a potential binding site for proteins involved in the autophagy/mitophagy process [65,66].

The effects of ischemia-reperfusion injuries on mouse models, in which 30 min of bilateral kidney ischemia is followed by 24 h of reperfusion, have been controversial. Even though a study revealed that such an injury results in an increase in the autophagy process characterized by the number of engulfed mitochondria at autophagosomes and the amount of degraded mitochondrial proteins on renal proximal tubular cells, another study demonstrated contradictory results claiming a decrease in autophagy and mitophagy [67,68,69]. The upregulation of mitophagy has been demonstrated in animal models of nephrotoxic agent-induced or sepsis-induced AKI [70,71]. Parkin-knockout mice models demonstrate worse histopathological and clinical outcomes following cisplatin or contrast medium exposure or sepsis [72].

The role of mitophagy in CKD has also been evaluated in multiple studies. Reductions in autophagic vesicle formation along with the downregulation of PINK1 and parkin expression have been demonstrated in diabetic kidney disease models on mice subjects [73,74]. Expression of another molecule, namely optineurin, which induces mitophagy and therefore leads to reductions in the formation of ROS and pro-inflammatory responses, is reduced in patients with diabetic kidney disease as shown from biopsy specimens [75]. Similarly, impairment of mitophagy has been illustrated in podocytes when exposed to a high glucose environment or in the specimens from mice models of streptozotocin-induced diabetic kidney disease [76,77]. Moreover, defective mitophagy has also been implicated in the pathophysiology of non-diabetic chronic kidney disease [78,79].

Even though mitophagy has been perceived as a protective mechanism by the removal of defective mitochondria without the release of pro-apoptotic or pro-inflammatory molecules including cytochrome c or mtDNA, it may become dysfunctional as the kidney disease progresses.

### 2.5. Oxidative Stress and Antioxidant Defense

Major antioxidant enzymes include superoxide dismutase (SOD), catalase, peroxiredoxin, and glutathione peroxidase [80]. Antioxidant enzymes are downregulated and inhibited in response to ischemia-reperfusion injury, as shown by a mouse model of kidney fibrosis, while supplementation with SOD mimetic agents from day 2 to 14 results in a reduction in kidney fibrosis [81]. Similar findings of a reduction in kidney fibrosis have been shown in animal models of cisplatin-induced AKI via SOD mimetic agents [82]. Additionally, excessive formation of ROS and defective antioxidant mechanisms also lead to structural damage to mitochondrial proteins and membranes causing the release of mitochondrial content into cytosol or further deterioration of antioxidant enzymes. Another antioxidant enzyme, referred to as catalase and located in peroxisomes, has shown to be reduced in AKI models via reduction in the number and function of peroxisomes, and it may be reversible via administration of proximal tubule-specific overexpression of NAD-dependent protein deacetylase SIRT1 [83,84]. Furthermore, cardiolipin are located at the inner mitochondrial membrane, preventing the release of cytochrome c into the cytosol which is peroxidated and becomes dysfunctional in response to oxidative stress [85]. A molecule that selectively binds to cardiolipin and prevents its peroxidation, namely szeto-schiller peptide 31 (SS-31), has shown to protect against various types of AKI and AKI-to-CKD transition [86,87].

In addition to their destructive roles, ROS formed within the mitochondria has the potential to regulate multiple signaling pathways, including the activation of the NLRP3-inflammasome pathway. This leads to an inflammatory response, including the release of cytokines and recruitment of pro-inflammatory immune cells to the tissue, the hypoxia-inducible factor 1α (HIF1α) leading to angiogenesis, and the transforming growth factor-β (TGFβ) pathway leading to tissue fibrosis [88,89,90].

### 2.6. Unfolded Protein Response

Mitochondrial proteins are encoded by both the nuclear and mitochondrial DNA and such proteins are prone to degeneration mainly due to oxidative stress. Similar to the cytosolic protein regulation system, the mitochondrial protein control system is dependent upon the chaperone proteins involved in the protein folding and proteases that remove undesired proteins [91]. Mutations at the TRAP1 gene encoding for a mitochondrial chaperone protein, which is highly expressed in proximal tubules and the thick ascending limb of Henle, is implicated in the pathophysiology of various congenital abnormalities of kidneys and urinary tracts, such as the VACTERL association [92]. Additionally, alterations in the expression of TRAP1 have been reported in animal models of AKI [93,94]. However, the studies investigating the role of mitochondrial unfolded protein responses in the pathophysiology of renal disorders are premature and the need for future studies is clear.

## 3. The Targeting of Mitochondria as a Therapeutic Approach in Kidney Disease

Mitochondrial dysfunction has been suggested to play a critical role in various forms of kidney injury and abnormal kidney repair. Several agents which target different mitochondrial processes to preserve and restore mitochondrial function as well as improve the treatment of patients with kidney diseases have recently emerged as potential therapeutic approaches in kidney pathology (Table 1, Figure 2 and Figure 3). According to their mechanisms, these compounds can be classified as cardiolipin protective agents, biogenesis activators, fission inhibitors, mPTP inhibitors, antioxidants, and drugs that target mitophagy. In addition, several glucose-lowering drugs, such as SGLT-2 inhibitors and GLP-1-RA, have been shown to act on mitochondrial processes.

### 3.1. Cardiolipin Protective Agents

Cardiolipin is a phospholipid that is present in the inner mitochondrial membrane and required for the optimal function of the mitochondria. Damage or deficiency of cardiolipin can result in altered membrane structure, stability and inadequate respiration. It leads to the release of cytochrome c and ROS generation resulting in the apoptosis of cells [3]. Structural alterations can release cardiolipins into the extracellular space and this can trigger the innate immune system via NOD-like receptors and inflammation pertaining renal injury and fibrosis [13]. Exposure to chronic hyperglycemia due to diabetes mellitus has been shown to decrease relative abundances of cardiolipins in cortical proximal tubules when compared with healthy controls [95] and depicts the role of cardiolipins in diabetic kidney disease. They are also involved in the progression of acute ischemic kidney injury and renal artery stenosis to CKD [87,96]. Therefore, targeting cardiolipins and preventing structural changes in mitochondria via cardiolipin protective agents can serve as a protection against mitochondrial insults in kidney cells as a result of diabetic kidney disease and ischemia.

#### 3.1.1. Szeto–Schiller Peptide 31 (SS-31)

SS peptides are mitochondria-targeting antioxidants which were found by Szeto and Schiller. Among these peptides, D-Arg-Dmt-Lys-Phe-NH2 (SS-31 aka MTP-131 or elamipretide or Bendavia) is the most studied in both preclinical and clinical settings [97]. SS-31 accumulates in the inner mitochondrial membrane binding cardiolipins and preventing peroxidation, structural damage, as well as electron leakage. It has no effect on healthy mitochondria [97]. In unilateral ureteral obstruction models, SS-31 was shown to counteract tubular apoptosis, oxidative stress and kidney damage [98,99]. In sepsis-induced AKI mice models, SS-31 treatment is associated with reductions in serum creatinine, BUN, AMPK, PGC-1a, and cleaved caspase-3 protein expression, indicating decreased apoptosis and maintained mitochondrial structure [100]. In addition to its direct mitochondrial effects, SS-31 was shown to inhibit aminopeptidase A, which converts angiotensin II to III, and to increase angiotensin II receptor type 2 expression which could limit kidney injury [101]. SS-31 was also shown to protect against diabetic kidney disease via maintaining physiologic superoxide levels [102], preventing tubulointerstitial injury by decreasing DRP1 expression and inhibiting mitochondrial fission [103], preventing cellular apoptosis [104] and ROS generation [105]. In renovascular disease models, SS-31 administration improved renal oxygenation, function, and hemodynamics while decreasing microvascular loss in stenotic kidneys [106]. SS-31 was also studied in metabolic syndrome models. In one study, glomerular size, glomerular injury, podocyte injury, impaired podocyte mitochondria, and foot process width were all increased with metabolic syndrome but restored with SS-31 administration [107]. In another study, an improvement in renal microvascular density and a decrease in microvascular remodeling was seen in pig models with metabolic syndrome after SS-31, suggesting a potential mitochondrial origin of intrarenal microvascular disease induced by metabolic syndrome [108]. Lipotoxicity to renal glomerules after high-fat diet was also suggested to be overcome by SS-31 administration via mitochondrial protection in mice models [109]. SS-31 was shown to prevent AKI following ischemia-reperfusion injury in mice models [110,111] and it was affiliated with a delay in CKD progression [13,87,112]. Lastly, it was suggested to prevent contrast-induced nephropathy in hypercholesterolemia mice [113] and cisplatin-induced AKI in mice models [114].

SS-31 has shown promising results in several preclinical kidney disease models; however, its transition into the clinics has been limited. One important barrier for the clinical use of SS-31 is its peptide structure. The difficulty to use peptides (i.e., SS-31) for chronic treatment could pose a significant limitation as peptides may not be directly suitable for use as convenient therapeutics as they potentially possess several intrinsic weaknesses, such as poor chemical and physical stability and a short circulating plasma half-life [115]. Nevertheless, these aspects need to be addressed for their use as clinical drugs [115].

Further models are required to better understand the underlying mechanisms, signaling pathways, and usefulness of SS-31, as well as to overcome its limitations in restoration of mitochondrial function and prevention of kidney disease.

There are currently 25 clinical trials on SS-31, among which two have been studied on kidney disease patients (NCT01755858, NCT02436447). In a pilot phase 2a clinical trial (NCT01755858), co-administration of SS-31 with percutaneous transluminal renal angioplasty was found to attenuate post-procedural hypoxia and to increase blood flow and kidney function [116]. However, the results of the other study have not been reported. Given these findings SS-31 is a suitable molecule in kidney disease patients; however, further pre-clinical and large-scale clinical studies are needed to overcome its certain drawbacks.

#### 3.1.2. Szeto–Schiller Peptide 20 (SS-20)

SS-20 is another SS peptide that can preserve mitochondrial structure via cardiolipin attachment but does not have ROS scavenging capacity and cannot inhibit mitochondrial permeability transition as SS-31 can. This molecule was studied to enhance the ischemia tolerance of the kidney. Pretreatment of rats before warm ischemia with SS-20 increased ischemia tolerance and reduced cellular swelling and breakdown, suggesting ongoing mitochondrial ATP production during the ischemic period. In addition to this, treatment after ischemia reduced fibrosis suggesting SS-20 as a promising agent to minimize both acute and chronic kidney disease following surgeries involving warm ischemia [117].

As with SS-31, SS-20 is also a peptide molecule and therefore, has the same disadvantages as SS-31, if transitioned into the clinics. There are currently no available clinical trials regarding SS-20 and further studies are required to better delineate this molecule’s role and benefits in kidney disease.

### 3.2. Fission Inhibitors

As a dynamic organelle, mitochondria can undergo fusion or fission depending on the environmental stimuli and stress to maintain an adequate mitochondrial population in the cell. Kidney diseases lead to an imbalance between these two processes, mostly fission dominating over fusion, and result in mitochondrial fragmentation [7]. Mitochondrial fission has been shown to be upregulated during podocyte injury in diabetic kidney disease [118], ischemia-reperfusion AKI [119], ischemic AKI [12,120], cisplatin-induced AKI [121], and septic AKI [122]. Excessive fission had also been shown to be related to fibrogenesis and play a role in obstructive nephropathy in human and mice renal fibroblast models [123]. This is why inhibition of fission via targeting the key protein DRP1 by mitochondrial division inhibitor-1 (mdivi-1), has been studied on several cellular and animal models.

#### Mitochondrial Division Inhibitor-1

Mdivi-1 is a quinazolinone which selectively inhibits dynamin-related protein 1 (DRP1) GTPase. This favors mitochondrial fusion instead of fission which plays a pathological role in both AKI and CKD. Although there is no clinical trial on this inhibitor, there are several reports about its use in cellular and animal models with kidney injuries. It was used in septic AKI cellular and animal models and reduced activation of NLRP3 mediated pyroptosis and improved mitochondrial function [124]. In septic rat models, mdivi-1 administration improved not only kidney but multi-organ function in response to endotoxin-mediated injury [125]. Mitochondrial fragmentation, apoptosis, and acute tubular injury of proximal tubular epithelial cells in response to cisplatin and ischemia/reperfusion were able to be alleviated by mdivi-1 in an in vivo study [11]. It was also shown to improve cardiac dysfunction due to renal ischemia/reperfusion in acute cardiorenal syndrome in mice models [126]. Mdivi-1 use was also investigated in CKD models. In a study on Pkd1-mutated mice, a model of polycystic kidney disease, mdivi-1 administration was shown to improve renal function and reduce kidney–body weight ratio and cyst formation [55]. This suggested a potential role of mitochondrial fission in polycystic kidney disease progression. Pharmacologic inhibition of DRP1 was successful in blocking mitochondrial fission in diabetic mice and blunted progression of diabetic nephropathy as a result [127]. In mouse models of obstructive nephropathy, mdivi-1 use was able to attenuate established renal fibrosis via a reduction in H3K27ac levels, fibroblasts, and interstitial fibrosis. This showed its potential role in epigenetic inhibition of fibrosis and a target to retard the progression of chronic kidney disease [123]. In contrast, mdivi-1 administration activated TGFB1-Smad2/3 signaling and worsened renal fibrosis after unilateral ureteral obstruction [78]. Mdivi-1 has shown promise in pre-clinical models; however, more evidence is needed to obtain concrete results about the use of mdivi-1 in kidney disease.

### 3.3. mPTP Inhibitors

A mitochondrial permeability transition pore (mPTP) is present in the inner mitochondrial membrane. Various matrices and inner and outer membrane proteins have been suggested as components of mPTP, such as the adenine nucleotide transporter, the CsA binding protein cyclophilin D, the voltage-dependent anion channel, hexokinase, aspartate-glutamate and phosphate carriers, and the spastic paraplegia 7 protein [128,129,130,131]. The opening of high-conductance mPTP as a result of cellular swelling, intracellular calcium overload after ischemic reperfusion, and cellular dysfunction leads to the release of pro-apoptotic molecules, such as cytochrome c, inducing renal cell apoptosis [3]. This process has been suggested to play a role in podocyte injury, ischemia/reperfusion injury, cisplatin nephrotoxicity [132], contrast-induced nephropathy [133], diabetic kidney disease [134], lead toxicity [135], and rhabdomyolysis related kidney injury [136]. There are several drugs that are used to inhibit mPTP, such as cyclosporine A, TDZD-8, and alisporivir.

#### 3.3.1. Cyclosporine-A

Cyclosporine-A is a calcineurin inhibitor used primarily as an immunosuppressant agent. At high doses, it is famously known as a nephrotoxic agent via inducing apoptosis and favoring mitochondrial fission in renal tubular cells [137]. However, in non-toxic doses, it has been used as an inhibitor of mPTP in several preclinical studies. It was shown to be protective against adriamycin-induced podocyte damage, decrease proteinuria, and restore mitochondrial morphology in rat models [138]. When administered via a 3 mg/kg i.v. dose in ischemia/reperfusion rat models, it was able to attenuate oxidative stress in rat kidneys and the study concluded its potential use as preconditioning against ischemia/reperfusion injury [139].

#### 3.3.2. TDZD-8

Glycogen synthase kinase (GSK) 3b is a constitutively active serine-threonine kinase that is present in all cell types, and it transfers regulatory signals downstream to modify mitochondrial permeability transition [140]. That is why inhibition of GSK3b by a highly selective inhibitor 4-benzyl-2-methyl-1,2,4-thiadiazolidine-3,5-dione (TDZD-8) can serve to inhibit mPTP and renal cell apoptosis. Its use improved proteinuria and glomerular sclerosis in adriamycin nephropathy mice models [140]. When given before or after ischemia/reperfusion, it suppressed renal fibrosis by reducing myofibroblasts, inflammatory cytokines, macrophages, and extracellular matrix deposition [141]. There is another study regarding its favorable effects on ameliorating non-steroidal anti-inflammatory drug (NSAID)-induced AKI via induction of renal cortical cyclooxygenase-2 (COX-2) and direct inhibition of mPTP [142].

#### 3.3.3. Alisporivir

Alisporivir, which is a non-immunosuppressive analogue of cyclosporine, is another inhibitor of cyclophilin and mPTP. It is not used as often as other inhibitors according to the literature. There is a report investigating its use in diabetic mice models showing no change in albuminuria and glomerulosclerosis with cyclophilin inhibition, which may exhibit the discredited role of mPTP in diabetic kidney disease [143]. There is one trial assessing the pharmacokinetics and safety of alisporivir in patients with end stage renal disease on hemodialysis (NCT01975337). However, the results of this study have not been reported.

### 3.4. Biogenesis Activators

Mitochondrial biogenesis is the action of increasing mitochondrial mass and mtDNA replication to meet the increase in demands in conditions such as AKI and CKD. The process is governed by the AMPK/SIRT/PGC1a pathway and activators of biogenesis can act as nephroprotective agents against both for AKI and CKD [16].

#### 3.4.1. SRT1720

SRT1720 is an activator of SIRT1 which is a key mediator of mitochondrial biogenesis. Administration of SRT1720 before unilateral ureteral obstruction partially attenuates renal fibrosis, oxidative stress, and apoptosis [144]. This indicates the possible use of SRT1720 in slowing CKD progression. However, in another study, its use increased renal fibrogenesis and alarmed a potential risk of progressive CKD in long-term use [145]. It was also suggested to be protective against cisplatin-induced AKI in mice [146]. SRT1720 administration during reperfusion for ischemic kidneys attenuated a potential AKI in rat models by allowing faster proximal tubule recovery [147,148]. Therefore, SRT1720 has several contradictory findings and may not be suitable for use in kidney disease. There are currently no available clinical trials regarding this molecule. Additional studies are needed to better understand its effects.

#### 3.4.2. LY344864

LY344864 is a selective 5-HT1F (5-hydroxytryptamine 1F) receptor agonist. It has been shown to activate mitochondrial biogenesis via PGC-1a activation and the simultaneous inhibition of biogenesis inhibitory pathways [149]. In ischemia/reperfusion-induced AKI mice, 5-HT1F agonism was responsible of accelerating renal function recovery and increased mtDNA copy numbers [150]. Similar to SRT1720, there are no available human studies for this molecule.

#### 3.4.3. BF175

BF175 is a potent and selective SIRT1 agonist. In cultured podocytes in a study, its use protected against hyperglycemia mediated mitochondrial injury in vitro. In the same study, in vivo experiments showed a marked reduction in albuminuria and glomerular injury in diabetic mice models [151]. This molecule was also offered as a potential drug for human immunodeficiency virus (HIV)-associated kidney disease [152]. In line with the other biogenesis activators SRT1720 and LY344864, BF175 has not been studied in human subjects.

#### 3.4.4. Resveratrol

Resveratrol is a plant-based substance abundantly found in grape skins and is reported to be beneficial for cardiovascular and renal diseases. It has antioxidant properties, scavenges free radicals, and enhances mitochondrial biogenesis through AMPK/SIRT1 and PGC1a activation. When it was given to diabetic mice for eight weeks, it attenuated renal pathological changes and partially improved diabetic and lipid laboratory parameters which indicates a better glucose-lipid metabolism [153]. Another study linked this improvement to its ability to reduce hyperglycemia-induced oxidative stress [154]. Its use in acute states, such as treatment during resuscitation to prevent potential ischemic/reperfusion AKI, showed an improved mitochondrial respiratory capacity, decreased mitochondrial ROS, and lipid peroxidation [155]. Resveratrol has been extensively investigated in clinical trials with 192 trials studying its role in various disease states. There are currently six studies regarding the role of resveratrol in kidney disease (NCT02433925, NCT03352895, NCT03597568, NCT03815786, NCT02704494, NCT03946176). One study (NCT02433925) investigated the role of resveratrol on inflammation and oxidative stress in individuals with CKD on conservative treatment strategies. Another study (NCT03352895) evaluated the effects of resveratrol on hearing impairment in patients with CKD on hemodialysis. However, among the given trials, only Sattarinezhad et al. (NCT02704494) [156] have reported their results which potentially indicates an absence of positive results. Sattarinezhad et al. showed that resveratrol may be an effective adjunct to angiotensin receptor blockers for a reduction in urinary albumin excretion in patients with diabetic nephropathy [156]. Although there are several promising findings regarding the effects of resveratrol, the sample sizes of the given clinical trials are limited, and the results of several trials are lacking. Therefore, future large-scale clinical studies are needed to further understand the benefits resveratrol in kidney disease patients.

#### 3.4.5. Formoterol

Formoterol is a long-acting b2 adrenergic receptor agonist which is primarily used as a bronchodilator for diseases including asthma and chronic obstructive pulmonary disease. In addition, it can increase PGC-1a synthesis and mitochondrial biogenesis. B2 adrenergic receptors are present on podocyte surfaces so formoterol can potentially be used in podocytopathies [157]. It has been shown to reverse hyperglycemia-induced changes in proximal tubular epithelial cells [158] and ischemia/reperfusion-induced AKI [159,160].

#### 3.4.6. Fibrates

Fibrates, including benzofibrate and fenofibrate, are primarily used as lipid-lowering drugs. They exert their effects through PPAR-alpha activation and increase mitochondrial biogenesis. Benzofibrate pretreated mice had significantly reduced cisplatin-induced renal necrosis and apoptosis [161]. Fibrates are also found to be beneficial in CKD via accelerating fatty acid oxidation. As most renal epithelial cells rely on fatty acids as an energy source, this prevents lipid accumulation, fibrosis, and CKD progression [162].

### 3.5. Antioxidants

Various mitochondria-targeting antioxidants have been investigated in different studies including MitoQ [73,163,164], Mito-CP [165], SkQ1 [7], SkQR1 [166], Ubiquinone (coenzyme Q10, CoQ10) [3,167], and Curcumin [168,169].

#### 3.5.1. MitoQ and Mito-CP

MitoQ is a ROS scavenger and antioxidant depositing at the matrix in a mitochondrial membrane potential-dependent mode [3]. MitoQ was shown to have a positive impact on injured renal tubules in diabetic kidney disease through the regulation of mitophagy by NF-E2-related factor 2 (Nrf2)/PTEN-induced putative kinase 1 (PINK) in diabetic mice [73]. MitoQ treatment was also reported to improve tubular and glomerular function with no considerable impact on plasma creatinine levels in a mouse model with type 1 diabetes [164]. In addition, both MitoQ and Mito-CP dose-dependently led to the prevention of cisplatin-induced kidney dysfunction in mice [165]. Mito-CP was also shown to prevent the dysfunction and injury of the mitochondria, inflammation in the kidneys, tubular injury, and apoptosis [165].

There are currently 26 trials on MitoQ, among which two involve kidney disease patients (NCT02364648, NCT03960073). Earlier trials with MitoQ involving Parkinson’s disease and multiple sclerosis did not report results, potentially suggesting, as in the case of resveratrol, an absence of positive results. NCT03960073 is investigating the effect of oxidative stress in the mitochondria on exercise capacity and arterial hemodynamics in patients with heart failure with preserved ejection fractions with and without CKD. This trial is still ongoing, whereas the other study (NCT02364648) has not reported its results. Given these findings, MitoQ has not shown significant promise regarding its role in the clinics. However, its preclinical results are somewhat promising and future studies are needed to better understand its effects in kidney disease.

#### 3.5.2. SkQs

The SkQs (plastoquinones), including SkQ1 and SkQR1, can accumulate in the mitochondria by penetrating planar phospholipid membranes [170,171]. SkQ1 resulted in a fall of renal oxidative stress and mitochondrial dysfunction in mitochondrial DNA mutator mice [171,172]. In addition, the administration of SkQR1 normalized ROS levels in kidney mitochondria, led to a fall in BUN and creatinine, and decreased disease-related mortality in experimental rats with ischemia/reperfusion injury compared to rats without treatment [166,171]. SkQR1 was also reported to be beneficial on gentamycin-induced nephrotoxicity [173]. The death of kidney epithelium cells, the severity of renal failure, and animal mortality caused by gentamycin application were reduced with SkQR1 treatment [173]. SkQR1 also decreased gentamycin-induced hearing loss [173].

Three trials have studied SkQ1 in dry-eye syndrome (NCT04206020, NCT03764735, NCT02121301) with promising results [174]. The effects of SkQ1 and SkQR1 in kidney disease patients requires further studies.

#### 3.5.3. Ubiquinone (Coenzyme Q10, CoQ10)

CoQ10 functions by normalizing ATP production, attenuating mitochondria reactive oxidative species, and decreasing mitochondrial membrane potential [3]. In one study, CoQ10 was administered as a therapeutic agent in a mouse model with type 2 diabetic nephropathy [167]. Absence in mitochondrial oxidized CoQ10 was shown to be a potential triggering factor for diabetic nephropathy and CoQ10 administration was reported to be potentially protective of kidney tissues in type 2 diabetes, via the conservation of the function of the mitochondria [167]. Multiple trials investigate the role of CoQ10 in kidney disease. Yeung et al. have performed a study in 20 participants to test whether CoQ10 therapy is safe, well-tolerated, and ameliorates biomarkers of oxidative stress in patients on hemodialysis (NCT00908297) [175]. They showed that CoQ10 administration at doses as great as 1800 mg per day was safe in all subjects and was well-tolerated in the majority. Short-term daily CoQ10 use lowered isofuran levels in the bloodstream in a dose-dependent manner and CoQ10 supplementation may potentially ameliorate the functioning of the mitochondria and lower oxidative stress in individuals on hemodialysis [175]. However, despite these findings, given the low sample size and the scarcity of clinical studies reporting concrete results surrounding the beneficial effects of CoQ10 in kidney disease patients, further pre-clinical and clinical studies are needed to investigate this molecule.

#### 3.5.4. Curcumin

Curcumin acts as an antioxidant and is responsible for altering mitochondrial dynamics and bioenergetics [3]. It was shown that curcumin can attenuate inflammation, renal fibrosis, and oxidative stress through the nuclear factor-erythroid-2-related factor (Nrf2) Kelch-like erythroid cell-derived protein with cap-n-collar homology-associated protein 1 (Keap1) pathway in rats subjected to 5/6 nephrectomy, suggesting its potential in the treatment of CKD [169]. Accordingly, in a separate study, curcumin administration in rats weakened oxidant stress and glomerular hemodynamical changes, such as hyperfiltration and hypertension, in the glomeruli induced by 5/6 nephrectomy [176]. These effects were related with the capabilities of this antioxidant to overturn renal structural changes, tubular atrophy, proteinuria, interstitial fibrosis, hypertension, fibrotic glomeruli, and expansion of the mesangium, further supporting the possibility that curcumin could be used in CKD by targeting established hemodynamical changes and renal injury [176]. Furthermore, curcumin has been shown to have strong antifibrotic effects in diabetic nephropathy and have renoprotective effects in mice mediated by inhibiting the NLRP3 inflammasome [168].

Curcumin has been investigated in a large number of trials. However, although it had promising preclinical data, the clinical effectiveness of native curcumin is weak [177].

Similar to curcumin, despite their efficiency and promising findings in experimental models of kidney dysfunction, the promise of mitochondria-targeted antioxidants as clinical therapeutic alternatives for kidney disease has not been promising and awaits further investigation.

Several problems exist surrounding the clinical translation of antioxidants [7]. In cell stress response pathways, low and moderate levels of mitochondrial ROS may function as signaling molecules [7,178]. Antioxidants can potentially have negative effects by scavenging and eliminating physiological levels of ROS that play a role in these signaling pathways [7]. The ROS levels measured prior to or during the supplementation of mitochondria-targeted antioxidants might potentially be beneficial in preventing the negative effects and allow for the use of these therapies with safety [7]. However, further developments are also required in these measurement methods and necessary equipment.

### 3.6. Enhancement of Mitophagy and Autophagy

The maintenance of mitochondrial quality control by mitophagy has protective functions during both AKI and CKD [179]. Therefore, enhancement of autophagy and/or mitophagy to remove damaged and dysfunctional mitochondria in kidney tubular cells has potential as a therapeutic strategy in kidney disease, especially in the prevention of AKI and diabetic kidney disease [7,179,180].

#### 3.6.1. mTOR Inhibitors

Mammalian target of rapamycin (mTOR) inhibitors are a possible candidate for this function as they have been reported to improve the elimination of damaged mitochondria in renal tissues through autophagy [181,182]. One study showed that rapamycin, an mTOR inhibitor, could ameliorate gentamicin-induced nephrotoxicity by enhancing autophagy [181]. Another study reported that rapamycin helped to protect against cisplatin-AKI by activating autophagy [182]. However, due to their immunosuppressive and antiproliferative effects, their use can be limited [7].

#### 3.6.2. Tat–Beclin 1

One potential inducer of autophagy is Tat–beclin 1, a derivative of beclin 1, an autophagy protein [183]. This molecule is an mTOR-independent agent and has been reported to potently stimulate autophagy in the pancreas, the skeletal and heart muscles, as well as the renal tubules in vivo [7,183].

#### 3.6.3. Urolithin A

Urolithin A, a gut microbial metabolite of ellagic acid and related compounds, was reported to provoke mitophagy both in vitro and in vivo following oral consumption [7,184]. Urolithin A is thought to activate SIRT1, SIRT3, and AMPK. Activated AMPK elevates PGC-1 levels which in turn directly increases mitochondrial biogenesis. The mTOR1 is inhibited by urolithin A and AMPK [185]. In Caenorhabditis elegans, urolithin A was responsible for the prevention of the buildup of dysfunctional mitochondria with age and prolonged lifespan, whereas in rodents urolithin A improved exercise capacity [184]. Oral administration of nanoparticle urolithin A was shown to normalize cellular stress and improve survival in a mouse model of cisplatin-induced AKI; however, the role of mitophagy in this kidney protection was not established [7,186].

There are currently eight trials investigating the role of urolithin A, none of which studied this molecule’s role in kidney disease patients. Liu et al. (NCT03283462) assessed whether urolithin A ameliorated the six minute walk distance, muscle stamina in hand and leg muscles, and biomarkers related with the health of the cell as well as mitochondria in adults aged 65 to 90 years [187]. They showed that even though ameliorations in the six minute walk distance and maximum ATP generation in the hand muscle were not significant in the urolithin A group in comparison with the placebo, long-term urolithin A supply had a positive effect on muscle endurance and the biomarkers in the bloodstream [187]. Despite the absence of clinical trials in kidney disease patients, the study conducted by Liu et al. is important as it shows the beneficial effects of urolithin A through mitochondrial pathways. Both pre-clinical and clinical studies are needed to further understand the potential roles of urolithin A in kidney disease patients.

#### 3.6.4. Other Strategies

The role of mitophagy in sepsis-induced AKI was also investigated [72,188]. One study showed that bone marrow-derived mesenchymal stem cells stimulated mitophagy and resulted in the inhibition of apoptosis as well as pyroptosis of renal tubular epithelial cells in kidney tissues by increasing the generation of SIRT1/Parkin, leading to the amelioration of sepsis-induced AKI [188]. Another study reported that polydatin was responsible in alleviating mitochondrial dysfunction in rat models of sepsis-induced AKI via Parkin-mediated mitophagy [72]. Furthermore, one study reported that melatonin played a protective role in diabetic nephropathy through the AMPK-PINK1-mitophagy pathway in mice [189]. Of note, there are conflicting results on the role of autophagy in kidney fibrosis [190,191].

### 3.7. The Role of Mitochondria in the Treatment of Polycystic Kidney Disease

Autosomal dominant polycystic kidney disease (ADPKD) is an important cause of CKD and is the most common single-gene inherited kidney disease in the world [192]. ADPKD results from mutated polycystic kidney disease 1 (PKD1) and PKD2, the genes, respectively, responsible for polycystin 1 and 2 [193]. Polycystins are reported to be of impact in the regulation of the pathways and molecules associated with energy generation and consumption, such as PPARα, calcium signaling in mitochondria-related membranes, PGC1α, cAMP, AMPK, mammalian targets of rapamycin complex 1 (mTORC1), and cystic fibrosis transmembrane conductance regulator (CFTR)-related ion transport [193]. Increased mTORC1 and a fall in AMPK signaling has played a role in cyst growth in ADPKD [192]. In several experimental studies, mTORC1 inhibition was shown to be effective [194,195]. However, there were contradictory results as clinical trials of mTORC1 inhibitors in ADPKD were not in parallel with preclinical studies as there was lower efficacy in actual patients [196,197]. The activation of AMPK is thought to alleviate cystic kidney disease severity by ameliorating mitochondrial biogenesis and decreasing the inflammatory mechanisms in tissues in animal models [192]. Metformin and canagliflozin are two of the agents that can activate AMPK [198,199]. In addition, several small molecules have been described to activate AMPK [192]. Favorable preclinical studies of AMPK activation to be used therapeutically in ADPKD are still ongoing [192]. In addition, as previously mentioned, one study reported that the use of Mdivi-1, which hinders with DRP1 function, significantly decreased the ratio between weight of the kidneys and the body, cyst generation, and ameliorated renal function in an orthologous mouse model with polycystic kidney disease [55]. Further studies are required to have therapies which successfully translate into the clinics.

### 3.8. Glucose-Lowering Medications

Aberrant mitochondrial function and the Warburg effect play crucial roles in the pathogenesis of diabetic kidney disease [8,200]. Some glucose-lowering agents have protective effects on the kidneys that could potentially be intrinsically related to the drug and not be related to their effects on glucose levels. GLP-1-RA and SGLT-2 inhibitors have protective effects on both cardiac and kidney tissues in individuals with diabetes [8]. Preclinical studies have suggested that the effect of these drugs on mitochondrial function may have contributed to their renal and cardiac benefits.

#### 3.8.1. GLP-1-RA

In addition to their positive impact on the mitochondria in other organs, such as the pancreas, brain, heart, liver, and retina, GLP-1-RA also improves mitochondrial function in the kidney [8].

In one study, high-fat-diet-induced obesity and CKD-positive rats had multiple metabolic conditions, including dyslipidemia, elevated body and renal weight, dysfunctional renal tissues, and glomerulosclerosis, as well as albuminuria and interstitial fibrosis [201]. Administration of liraglutide significantly ameliorated these metabolic derangements and lessened the level of injured mitochondria. This agent also had a beneficial effect on mitochondrial metabolites, such as fumarate, succinate, NAD+, citrate, and taurine in the kidney and helped to restore mitochondrial function in the kidneys to some extent via the SIRT1/AMPK/PGC1α pathways [201]. However, it is unclear if the improvements in mitochondrial function were a directly a result of liraglutide or secondary to a better metabolic profile.

GLP-1-RA have also been reported to activate SIRT1. SIRT1 has antifibrotic, antiapoptotic, blood pressure-regulating, and anti-inflammatory roles [202]. SIRT1 was shown to be decreased in the kidneys of streptozotocin-induced diabetic mice [203]. In addition, SIRT1 insufficiency has been related to renal lesions caused by aging, renal interstitial fibrosis, tubular cell apoptosis, and renal hypoxia [204]. Thus, the activation of SIRT1 by GLP-1RA may be contributing to the kidney protective effects of these medications in diabetic kidney disease [8].

GLP-1-RA were also shown to have effects on autophagy and oxidative stress. Liraglutide stimulated autophagy and reduced oxidative stress in diabetic kidney disease rats [205]. The protective effects on the kidney of GLP-1 came via the GLP-1R-AMPK-mTOR-autophagy-ROS signaling axis [205]. In addition, GLP-1-RA were reported to increase mitophagy, autophagy, and mitochondrial biogenesis to minimize the remodeling of the heart [206].

#### 3.8.2. SGLT-2 Inhibitors

SGLT-2 inhibitors have also been suggested to have beneficial effects on mitochondrial function. SGLT-2 inhibitors stimulate mitochondrial biogenesis in the cardiac as well as renal tissues through SIRT1 signaling [8]. Hypoxia-inducible factor (HIF)-2α, once activated by SGLT-2 inhibitors, enhance autophagy and mitophagy. Indeed, a fall in SIRT1, AMPK, and HIF-2α in different diseases such as obesity, CKD, type 2 diabetes mellitus, and chronic heart failure results in these molecules becoming appealing as potential targets to be used in therapy [8,207].

One study showed that glucose in contact with proximal tubule cells, mesangial cells, and podocytes led to Bax being translocated to the mitochondria and an elevated apoptotic index, as well as a fall in BCL-2 in the proximal tubular and cells in the mesengium [208]. Dapagliflozin administration led to SGLT-2 downregulation and eradicated the apoptotic response, suggesting that SGLT-2 inhibitors may have beneficial effects on mitochondrial metabolism [208]. SGLT-2 inhibitors also promote the fusion and fission of mitochondria [8].

SGLT-2 inhibitors also decrease oxidative stress. In type 1 diabetic mice, dapagliflozin suppressed ROS production in the kidney, as detected by dihydroethidium [209]. Thus, dapagliflozin was shown to ameliorate diabetic nephropathy by suppressing oxidative stress resulting from elevated glucose by a mechanism independent of glycemic control in mice [209].

#### 3.8.3. Other Glucose-Lowering Medications

Other glucose-lowering agents also have some mitochondrial effects. The buildup of metformin in mitochondria via its positive charge and inhibition of complex I of the respiratory chain (NADH dehydrogenase or NADH: ubiquinone oxidoreductase) is assumed to contribute to AMPK activation [210]. Mitochondrial respiration is decreased, and ATP synthesis is impaired following complex I inhibition. This can result in a rise in ADP/ATP and AMP/ATP ratios which trigger AMPK [8]. AMPK activation, in turn, improves the biogenesis of the mitochondria and autophagy [8].

In addition, metformin inhibits mitochondrial glycerophosphate dehydrogenase (mGPDH), an enzyme in the redox shuttle, in a non-competitive matter [211]. This inhibition potentially alters the hepatocellular redox state, reduces transformation of lactate and glycerol to glucose, and decreases hepatic gluconeogenesis [211]. Metformin also possibly attenuates the development of atherosclerosis via reductions in DRP1-mediated mitochondrial fission in an AMPK-dependent manner [212].

Thiazolidinediones, such as rosiglitazone and pioglitazone, act on PPARγ and in turn activate PGC1α [213]. In addition, thiazolidinediones play a role in activating PPARγ-independently from AMPK signaling [7]. Several studies have reported the kidney protection effects of thiazolidinediones in both animal models of AKI and diabetic kidney disease, as well as in patients with diabetic kidney disease [7,214,215,216,217]. However, the role of mitochondrial pathways such as mitochondrial biogenesis in the renoprotective effects of thiazolidinediones requires further investigation [7].

Dipeptidyl peptidase 4 (DPP-4) inhibitors also have beneficial effects on mitochondrial functions. These can potentially be related to their glucagon-like peptide-1 receptor engagement [8].

## 4. Conclusions, Future Directions, and Perspectives

In this review, we discussed the importance of mitochondrial biology and mitochondria-targeted therapies in kidney disease. Even though mitochondria-focused therapeutic alternatives have only been investigated for a short period of time, there are multiple ongoing promising clinical trials examining their effects on short- and long-term follow-up. In addition to the clinical trials studying the effects of certain chemicals and drugs in kidney disease, there are several additional trials investigating mitochondrial dysfunction in CKD (NCT02923063), AKI (NCT05264285, NCT05458063), diabetic kidney disease (NCT04074668, NCT05319990, NCT05071287), and PKD (NCT04630613). Novel therapeutic approaches include cationic mitochondrial penetrating peptides, such as SS-31, nanocarriers, and nanoceria, which are cerium oxide nanoparticles that have antioxidant capacities on their own and shown to improve renal function in AKI models and mitochondrial transplantation [218,219]. Mitochondrial transplantation is a new experimental technique that aims to transplant healthy mitochondria into the target tissue for treatment. Mitochondrial transplantation therapy has shown to increase renal function and antioxidant enzyme levels in an AKI model [220,221]. In addition, it increased the proliferation of renal cells and reduced inflammation [221]. However, there is clear need for large-scale clinical trials for the assessment of mitochondria-targeted therapy in the treatment of kidney disease. Furthermore, additional studies are required to better understand the pathophysiological role of the mitochondria in kidney disease and to provide an improved approach to translate these findings into clinics.

## Figures and Tables

**Figure 1 pharmaceutics-15-00570-f001:**
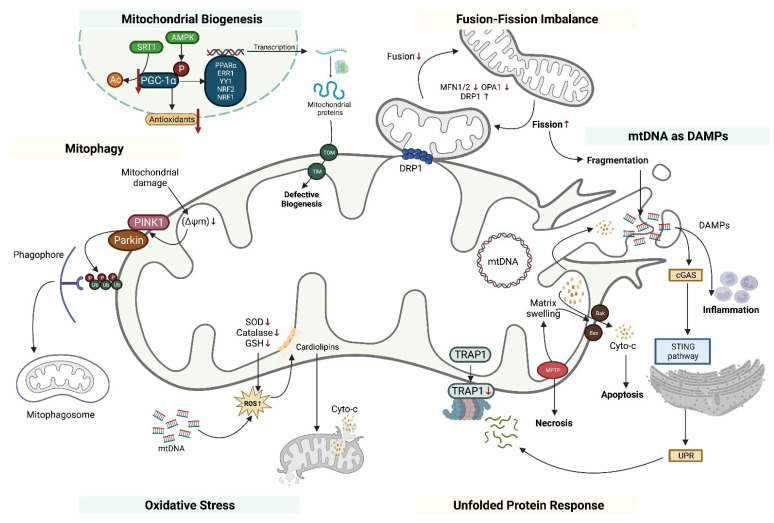
The pathophysiological pathways in mitochondrial dysfunction in kidney diseases including abnormal mitochondrial biogenesis, fusion-fission imbalance, mitochondrial DNA (mtDNA) acting as damage-associated molecular patterns (DAMPs), unfolded protein response, oxidative stress, and abnormal mitophagy. PTEN, induced kinase 1; mtDNA, mitochondrial DNA; SOD, superoxide dismutase; GSH, glutathione; SRT1, sirtuin 1; PGC-1α, peroxisome proliferator-activated receptor gamma co-activator 1 alpha; PPAR-α, peroxisome proliferator-activated receptor alpha; ERR1, estrogen-related receptor 1; NRF, nuclear respiratory factor; MFN, mitofusin; OPA1, protein optic atrophy 1; DRP1, dynamin-related protein 1; UPR, unfolded protein response; DAMP, damage-associated molecular patterns; cGAS, cyclic GMP-AMP synthase; ROS, reactive oxygen species; Cyto c, cytochrome c; and TRAP1, tumor necrosis factor receptor-associated protein 1. Upward pointing arrows indicate “increase”; downward arrows indicate “decrease”.

**Figure 2 pharmaceutics-15-00570-f002:**
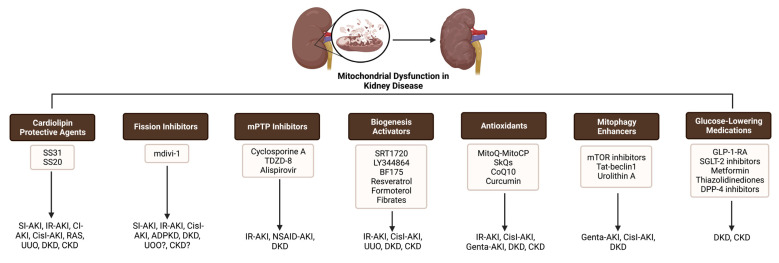
Drugs that target dysfunctional mitochondrial processes in kidney diseases. Szeto–Schiller peptide 20, SS-31: Szeto–Schiller peptide 31, Mdivi-1: mitochondrial division inhibitor 1, TDZD-8: 4-benzyl-2-methyl-1,2,4-thiadiazolidine-3,5-dione, SkQs: plastoquinone, CoQ10: coenzyme Q10, mTOR: mammalian target of rapamycin, GLP-1-RA: glucagon-like peptide-1 receptor agonists, SGLT-2: sodium glucose co-transporter-2, DPP-4: dipeptidyl peptidase 4, SI-AKI: sepsis-induced acute kidney injury, IR-AKI: ischemia/reperfusion-induced acute kidney injury, CisI-AKI: cisplatin-induced acute kidney injury, CI-AKI: contrast-induced acute kidney injury, NSAID-AKI: non-steroidal anti-inflammatory drug-induced acute kidney injury, Genta-AKI: gentamicin-induced acute kidney injury, RAS: renal artery stenosis, UUO: unilateral ureteral obstruction, DKD: diabetic kidney disease, CKD: chronic kidney disease, ADPKD: autosomal dominant polycystic kidney disease.

**Figure 3 pharmaceutics-15-00570-f003:**
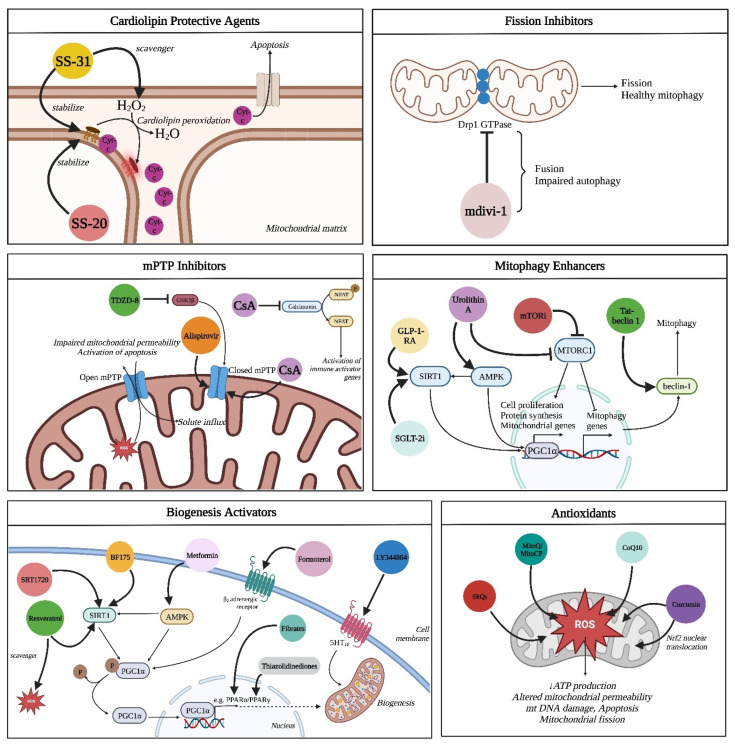
Mechanism of actions of mitochondria-targeting drugs that are used in kidney diseases. cyt-c: cytochrome c, SS-31: Szeto–Schiller peptide 31, SS-20: Szeto–Schiller peptide 20, DRP1 GTPase: dynamin-related protein 1 guanosine triphosphatase, mdivi-1: mitochondrial division inhibitor 1, GSK3b: glycogen synthase kinase 3beta, NFAT: nuclear factor of activated T-cells, ROS: reactive oxygen species, mPTP: mitochondrial permeability transition pore, CsA: cyclosporine A, SIRT1: Sirtuin1, AMPK: AMP-activated protein kinase, MTORC1: mammalian target of rapamycin complex 1, PGC1a: peroxisome proliferator-activated receptor-gamma coactivator-1a, mTORi: mammalian target of rapamycin inhibitor, PPARa: peroxisome proliferator-activated receptor alpha, 5HT1F: 5-hydroxytryptamine receptor 1F, SkQs: plastoquinone, CoQ10: coenzyme Q10, Nrf2: the nuclear factor erythroid 2-related factor 2, ATP: adenosine triphosphate, mtDNA: mitochondrial DNA, GLP-1-RA: glucagon-like peptide-1 receptor agonists, SGLT-2i: sodium glucose co-transporter-2 inhibitors. Downward arrow indicates “decrease”.

**Table 1 pharmaceutics-15-00570-t001:** Mitochondria-targeting drugs that are used in kidney diseases.

Drug Name	MECHANISM of Action	Animal Models	Clinical Trials on Kidney Disease Identifier Number	Kidney Diseases Studied on Humans
**Cardiolipin Protective Agents**				
SS-31	Binds cardiolipins, prevents their peroxidation, and maintains mitochondrial membrane structure, ROS scavenger	Unilateral ureteral obstruction (98, 99), sepsis-induced acute kidney injury (100), diabetic kidney disease (102–105), renovascular disease (106), metabolic syndrome (107–109), ischemia-reperfusion acute kidney injury (110, 111), contrast-induced nephropathy (113), cisplatin-induced acute kidney injury (114), chronic kidney disease (13, 87, 112)	NCT01755858, NCT02436447	NCT02436447: patients with renal impairment NCT01755858: percutaneous transluminal renal angioplasty
SS-20	Binds cardiolipin and maintains mitochondrial membrane structure, but does not scavenger ROS and inhibit inner mitochondrial membrane transition	Warm ischemia kidney models and ischemia-induced fibrosis (117)	N/A	N/A
**Fission inhibitors**				
mdivi-1	Selectively inhibits DRP1 GTPase and favors mitochondrial fusion	Septic acute kidney injury (124, 125), cisplatin-induced acute kidney injury (11), ischemia reperfusion-induced acute kidney injury (11, 126), autosomal dominant polycystic kidney disease (55), chronic kidney disease (123), diabetic kidney disease (127), unilateral renal obstruction (78, 123)	N/A	N/A
**mPTP Inhibitors**				
TDZD-8	Inhibits glycogen synthase kinase 3b and prevents mPTP channel opening	Adriamycin-induced kidney damage (140), ischemia-reperfusion acute kidney injury (141), NSAID-induced acute kidney injury (142)	N/A	N/A
Alisporivir	Analogue of cyclosporine A. Binds cyclophilin and inhibits mPTP channels	Diabetic kidney disease (143)	NCT01975337	NCT01975337: hemodialysis patients
**Biogenesis Activators**				
SRT1720	Activates Sirtuin1 and mitochondrial biogenesis	Unilateral ureteral obstruction (144), chronic kidney disease (144, 145), cisplatin-induced acute kidney injury (146), ischemia-reperfusion acute kidney injury (147, 148)	N/A	N/A
LY344864	Selectively activates 5-HT1F receptor and activates mitochondrial biogenesis	Ischemia-reperfusion acute kidney injury (150)	N/A	N/A
BF175	Selectively activates Sirtuin1 and mitochondrial biogenesis	Diabetic kidney disease (151), HIV kidney disease (152)	N/A	N/A
Resveratrol	Activates mitochondrial biogenesis and ROS scavenger	Diabetic kidney disease (153, 154), ischemia-reperfusion acute kidney injury (155)	NCT02433925, NCT03352895, NCT03597568, NCT03815786, NCT02704494, NCT03946176	NCT02433925, NCT03352895, NCT03597568, NCT03815786: chronic kidney disease, NCT02704494: diabetic kidney disease, NCT03946176: hemodialysis patients
**Antioxidants**				
MitoQ	ROS scavenger and antioxidant which concentrates at the matrix in a mitochondrial membrane potential-dependent manner	Diabetic kidney disease (73, 164), cisplatin-induced acute kidney injury (165)	NCT02364648, NCT03960073	NCT02364648: chronic kidney disease NCT03960073: HFpEF patients with and without chronic kidney disease
SkQ	Accumulates in mitochondria by penetrating planar phospholipid membranes	Mitochondrial DNA mutator mice (172), ischemia/reperfusion injury (166), gentamycin-induced nephrotoxicity (173)	N/A	N/A
Ubiquinone (coenzyme Q10, CoQ10)	Normalizes ATP production, attenuates mitochondria reactive oxidative species and decreases mitochondrial membrane potential	Diabetic kidney disease (167)	NCT04445779, NCT05422534, NCT03579693, NCT04972552, NCT00307996, NCT00908297, NCT05266794, NCT05214885, NCT01408680	NCT04445779: acute kidney injury/failure NCT05422534: end stage renal disease NCT03579693: chronic kidney disease NCT04972552: kidney transplantation NCT00307996, NCT00908297, NCT05266794, NCT01408680 hemodialysis patients NCT05214885: clear cell renal cell carcinoma
Curcumin	Acts as an antioxidant and is responsible for altering mitochondrial dynamics and bioenergetics	5/6 nephrectomy rats (169, 176), diabetic kidney disease (168)	NCT02369549, NCT03475017, NCT04413266, NCT03223883, NCT03935958, NCT03262363, NCT02494141, NCT04132648, NCT01285375, NCT01831193, NCT01225094, NCT05183737, NCT03019848, NCT04890704, NCT01906840	NCT02369549, NCT03475017, NCT03223883, NCT03262363, NCT04132648, NCT05183737: chronic kidney disease NCT04413266: peritoneal dialysis NCT03935958, NCT01285375: kidney transplantation NCT02494141: autosomal dominant polycystic kidney, NCT01831193, NCT03019848: proteinuria NCT01225094, NCT04890704: acute kidney injury NCT01906840: end stage renal disease
**Enhancement of Mitophagy and Autophagy**				
Tat–beclin 1	Inducer of autophagy, derived from a region of the autophagy protein, beclin 1	Mice infected with chikungunya or West Nile virus (183)	N/A	N/A
Urolithin A	A gut microbial metabolite of ellagic acid and related compounds was reported to induce mitophagy	Cisplatin-induced AKI (186), two different mouse models of age-related decline in muscle function, as well as young rats and Caenorhabditis elegans (184)	N/A	N/A

AKI: acute kidney injury, ATP: adenosine triphosphate, DRP1 GTPase: dynamin-related protein 1 guanosine triphosphatase, HFpEF: heart failure with preserved ejection fraction, HIV: human immunodeficiency viruses, Mdivi-1: mitochondrial division inhibitor 1, MPTP: mitochondrial permeability transition pore, NSAID: non-steroidal anti-inflammatory drugs, ROS: reactive oxygen species, SkQs: plastoquinone, SS-20: Szeto–Schiller peptide 20, SS-31: Szeto–Schiller peptide 31, TDZD-8: 4-benzyl-2-methyl-1,2,4-thiadiazolidine-3,5-dione, 5HT1F: 5-hydroxytryptamine receptor 1F.

## Data Availability

Not applicable.

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
