# Peer review of "The Mitochondrion: A Promising Target for Kidney Disease"

_pharmaceutics, 2023, doi:10.3390/pharmaceutics15020570_

Round 1

Reviewer 1 Report (New Reviewer)

In this review the authors give an overview the role of mitochondria in acute and chronic kidney diseases, the molecular pathways involved in mitochondrial dysfunction and possible treatment targets. In detail they summarize the current state of clinical research in this regard. The amount of clinical trials clearly shows the relevance of this topic. 

The work is well-researched. They provide three figures with schemes and one table to support their text. The figures are well designed and helpful. 

I have a few suggestions to improve the review:

  1. The glucose lowering medications should appear in the figures.
  2. It is not very clear to my why polycystic kidney disease get an own chapter. This chapter seems a bit out of place. I recommend to place the information in the appropiate sections.
  3. Further comments and small corrections can be found as comments in the attached PDF.

Author Response

Reviewer 1.

In this review the authors give an overview the role of mitochondria in acute and chronic kidney diseases, the molecular pathways involved in mitochondrial dysfunction and possible treatment targets. In detail they summarize the current state of clinical research in this regard. The amount of clinical trials clearly shows the relevance of this topic.

The work is well-researched. They provide three figures with schemes and one table to support their text. The figures are well designed and helpful.

Response: We would like to thank our reviewer for their kind words as well as helpful and constructive comments, we truly appreciate their guidance.

I have a few suggestions to improve the review:

The glucose lowering medications should appear in the figures.

Response: We would like to thank our reviewer for mentioning this issue we revised our figures.

It is not very clear to my why polycystic kidney disease get an own chapter. This chapter seems a bit out of place. I recommend to place the information in the appropiate sections.

Response: Thank you for bringing this up. We wanted to provide a concrete overview of how mitochondrial pathophysiology and therapeutics can affect patients with PKD. Instead of providing an entirely new chapter we instead placed this section as a subheading under our therapeutics section. We wanted to preserve its compact nature and left it as a single section.

Further comments and small corrections can be found as comments in the attached PDF.

Response: Thank you, we revised our text accordingly.

Reviewer 2 Report (New Reviewer)

In the manuscript “The Mitochondrion: A Promising Target for Kidney Disease”, the authors explored the importance of mitochondrial dysfunctions in kidney disease development, pointing mitochondrial as an important therapeutic target. The text was well organized, table and figures illustrated the text suitably. However, some minor points could be reviewed, as follow:

Minor points: 

  • The English needs to improved by a native English Speaker.
  • The authors stated that: “The kidney has the second highest mitochondrial content and oxygen consumption after the heart”. Although renal cells are rich in mitochondria, liver and muscle have a massive amount. The author’s statement are over estimated and must include the references.
  • Table1: The table was well organized and presents essential informations . However, references must be incorporated.
  • Figure 2: Improve the figure legend and the image distribution.
  • Other important points of the mitochondria / renal disease relationship as the walburg effect, the modulation of mitochondrial functions by polycystins, glutamine and aminoacid metabolism and oxidative phosphorylation were not explored. As a review, these topics could significantly enrich the manuscript.

Author Response

Reviewer 2.

In the manuscript “The Mitochondrion: A Promising Target for Kidney Disease”, the authors explored the importance of mitochondrial dysfunctions in kidney disease development, pointing mitochondrial as an important therapeutic target. The text was well organized, table and figures illustrated the text suitably. However, some minor points could be reviewed, as follow

Response: We thank and truly appreciate our reviewers’ contributions in helping to improve our manuscript.

Minor points:

The English needs to improved by a native English Speaker.

Response: Thank you, we revised the grammar and vocabulary of our text substantially to make it more legible and understandable.

The authors stated that: “The kidney has the second highest mitochondrial content and oxygen consumption after the heart”. Although renal cells are rich in mitochondria, liver and muscle have a massive amount. The author’s statement are over estimated and must include the references.

Response: Thank you, this statement can, in fact, be found in a number previous studies, we have also added the references (https://doi.org/10.1038%2Fnrneph.2017.107)

Table1: The table was well organized and presents essential informations . However, references must be incorporated.

Response: Thank you, revised.

Figure 2: Improve the figure legend and the image distribution.

Other important points of the mitochondria / renal disease relationship as the walburg effect, the modulation of mitochondrial functions by polycystins, glutamine and aminoacid metabolism and oxidative phosphorylation were not explored. As a review, these topics could significantly enrich the manuscript.

Response: Thank you, revised.

Reviewer 3 Report (Previous Reviewer 2)

The revised version appears to be much improved. I feel that the authors have addressed my suggestions adequately.

Author Response

Reviewer 3.

The revised version appears to be much improved. I feel that the authors have addressed my suggestions adequately.

Response: We would like to thank our reviewer for their careful and thorough reading of this manuscript and for their previous thoughtful comments as well as their kind words, they have helped improve the quality of this manuscript.

This manuscript is a resubmission of an earlier submission. The following is a list of the peer review reports and author responses from that submission.

Round 1

Reviewer 1 Report

The present review addresses mitochondrial dysfunction and its related pathophysiological aspects in different forms of kidney disease. The authors review literature to provide evidence on the relationship between acute kidney injury (AKI) and chronic kidney disease (CKD) and alterations in mitochondrial biogenesis, imbalance between mitochondrial fission and fusion processes, oxidative stress and mitophagy. However, the review presents non-original content (PMDI: 34681922) and brings an extremely simplified view of the mechanisms recognized by its complex regulatory system, which can lead the reader to form concepts that are not compatible with the complexity of the systems.

Major points:

1) The authors describe the sites of formation of reactive oxygen species (ROS) on page 6, lines 227-229: “Electron transport chain proteins, especially the complex I and III, are the major sites for the formation of reactive oxygen species (ROS) through the interaction of leaked electrons and oxygen to form superoxide anion initially” First, there is a conceptual error, some sources of mitochondrial oxidants can directly release hydrogen peroxide (PMID: 28842493). In addition, the production of mitochondrial oxidants is typically attributed to the canonical components of the electron transport chain. Notably, complex I, complex II and the ubiquinone cycle within complex III are biologically-relevant sources of oxidants. But it is important not to limit mitochondrial oxidant production to the electron transport chain. There are mitochondrial enzymes capable of producing superoxide or hydrogen peroxide radicals as a by-product of their primary function, including glycerol 3 phosphate, pyruvate, alphaketoglutarate, dihydroorotate, and very long-chain acyl-CoA dehydrogenases, as well as electron transfer flavoprotein (PMID: 15356189, PMID: 15356188, PMID: 20064600, PMID: 28842493). Although these sources are often neglected as important sources of mitochondrial oxidants, under certain conditions the release of reactive intermediates from these enzymes can be significantly greater than that of components of the electron transport chain.

2) The authors describe the mitochondrial permeability transition pore (mPTP) on page 9, lines 383-385: “Mitochondrial permeability transition pore (mPTP) which is present in inner mitochondrial membrane consists of voltage dependent anion channel, adenine nucleotide transporter and cyclophilin D”. Various matrix, inner and outer membrane proteins have been suggested as components of mPTP, such as the adenine nucleotide transporter, the CsA binding protein cyclophilin D, the voltage-dependent anion channel, hexokinase, aspartate-glutamate and phosphate carriers and the spastic paraplegia 7 protein (PMID: 20026006, PMID: 9013575, PMID: 18667415, PMID: 26387735). There is still a large amount of data in the literature discussing the molecular identity of mPTP components. Therefore, for a good approach to mitophagy induced by the opening of the mitochondrial permeability transition pore, a good theoretical foundation is necessary.

Author Response

We would like to thank our reviewer for their helpful and constructive comments. This is an in-depth review of mitochondrial dysfunction and the role of mitochondria as a potential target for the treatment of kidney disease. Mitochondrial dysfunction is a critical topic in kidney disease and the related information is constantly expanding. Therefore, we wanted to present an up-to-date review.

 In comparison to other review articles, we provide, to the best of our knowledge, the most up to date information regarding both pre-clinical studies and clinical trials. We discuss most of the present drugs with their current clinical trials both related to kidney disease and, if these drugs have not been used in clinical kidney disease patients, we assess their benefits and use in other diseases, try to infer and draw a conclusion from their effects on other diseases and pre-clinical studies.

After presenting the pathophysiological mechanisms of mitochondrial dysfunction in kidney disease, twe discuss each drug in detail and bring a critique to most drugs in line with the available evidence and clinical trials. Finally, in our future perspectives section we present several additional trials which use interventions other than drugs. In addition, We’ve added a new table which explains the mechanisms of the drugs together with the available evidence derived from animal models and clinical trials.

We also provide 3 original figures for our review article.

Reviewer 2 Report

Review is very well written and appears to be quite exhaustive when discussing the sheer number of mitochondrial-related drugs which could be used to treat kidney disease.

Maybe should have another figure or table highlighting the effectiveness of some of these drugs to successfully treat kidney disease in animal models.

A few minor comments:

Line 25 – a double space is present.

Line 65 – not sure what the purpose of this line is. Maybe a heading for the next section.

Line 188 – Maybe change “Oxidative mtDNA …” to “Oxidized mtDNA …”

Line 503 – should have a period and space removed.

Author Response

Answer: We would like to thank our reviewer for their comments. We really appreciate your support and are grateful for your kind remarks. Thank you. We’ve added a new table which explains the mechanisms of the drugs together with the available evidence derived from animal models and clinical trials.

A few minor comments:

Line 25 – a double space is present.

Answer: Thank you, revised

Line 65 – not sure what the purpose of this line is. Maybe a heading for the next section.

Answer: Thank you for pointing this out. As you have stated, this is our heading for the next section. “Mitochondria as A Key Regulator of Kidney Disease: The Pathophysiological Basis”

Line 188 – Maybe change “Oxidative mtDNA …” to “Oxidized mtDNA …”

Answer: Thank you, revised

Line 503 – should have a period and space removed.

Answer: Thank you, we deleted this section as we had already discussed SS-31 elsewhere in the text.

Reviewer 3 Report

1. The introduction (lines 47-57) needs references.

2. The review is well written however, can the authors respond to how the information collected in this review is novel. Most of the concepts presented in the manuscript are readily available in existing review articles. 

Author Response

Thank you. The references are revised

We would like to thank our reviewer for their comment. This is an in-depth review of mitochondrial dysfunction in kidney disease. Mitochondrial dysfunction is a critical topic in kidney disease and the related information is constantly expanding. Therefore we wanted to present an up to date review of mitochondrial dysfunction in kidney disease. We initially explain the pathophysiological mechanisms of mitochondrial dysfunction in kidney disease, then we discuss each drug in detail and bring a critique to most drugs in line with the available evidence and clinical trials. Finally, in our future perspectives section we present several additional trials which use interventions other then drugs. We’ve added a new table which explains the mechanisms of the drugs together with the available evidence derived from animal models and clinical trials.

In comparison to other review articles, we provide, to the best of our knowledge, the most up to date information regarding both pre-clinical studies and clinical trials. We discuss most of the present drugs with their current clinical trials both related to kidney disease and, if these drugs have not been used in clinical kidney disease patients, we assess their benefits and use in other diseases, try to infer and draw a conclusion from their effects on other diseases and pre-clinical studies.

We also provide 3 original figures for this review article

Reviewer 4 Report

The review by Tanriover et all aims to explore if targeting mitochondria is a promising approach to treat kidney disease. Unfortunately, I left more confused after reading the manuscript than before. I fully acknowledge that mitochondria as pharmacological target is a difficult subject where multiple pathologies and signaling pathways are interlinked and cannot be independently interrogated. This, however, is even more reason to display a significant amount of scrutiny towards this topic. A good review should not be a sales pitch but a critical analysis of the literature. Unfortunately, in this regard this review falls short. I would have liked to read a more balanced account on the different pathologies and therapeutic strategies. Especially a clear separation of association and causation for each individual point made would have strengthened the review. In this process the review could have been shortened significantly, for example by relying on pure causations in the text and listing mere associations in tabular format.

In addition, the therapeutic options could have been approached more critical too. For example, but certainly not limited to, the difficulty to use peptides (i.e. SS31) for chronic treatment should be highlighted as significant limitation. Similarly, curcumin is by now a classic example of a substance with promising preclinical data in multiple indications that failed in every properly controlled and powered clinical trial. The same can be said for all clinically tested antioxidants at this point. Therefore, listing any compound in a review without a disclaimer provides the wrong perspective. This critical assessment of all therapeutic options is missing throughout the review. Clearly, clinical trials have been conducted and finalized with some of the molecules in question. Even if these compounds have been used in other indication, these trials could provide some information of what to expect (or not) and what limitations might be associated (i.e. Saad et al. 2017). For example, earlier trials with MitoQ did not report results (i.e. Parkinsons, MS) which likely indicates an absence of positive results, since positive results are typically published to attract investors. In this sense I was disappointed to see a list of clinical trials currently underway without interpretation of their potential to succeed. On the subject of clinical trials, the statement in the introduction that “…we highlight the current clinical trials…” is clearly not represented by the manuscript as apart from the simple listing of trial numbers in the last paragraph, even inclusion of clinical trials in the majority of the manuscript is extremely scarce. Similarly, listing clinical trials that will be unable to differentiate between the activity of a drug versus other forms of intervention (i.e. exercise NCT05422534, this trial has not started yet, therefore, to list it as ongoing is not correct) will further complicate discussion around what works and what does not.

In general, I believe this review is of importance but in its present form it does not provide the reader with sufficient scientific scrutiny to separate it from an enthusiastic sales pitch. One occurrence where the authors briefly approached a true critique was the comparison of cyclosporin with alisporivir and I applaud the authors for this inclusion. This critical assessment should have been extended to all listed potential therapeutic options.

Minor points

·         This manuscript should be edited by a native speaker to improve grammar.

·         I cannot find Trial NCT0292306, this is likely the result of a mistake (one digit is missing) but highlights that including and critically discussing the relevant trials into the manuscript for each section would improve the manuscript.

·         It is unclear why NCT04984226 was included as veverimer’s mode of action is to neutralise HCL in the gut without a direct mitochondrial association for kidney mitochondria (again, this could be discussed in the manuscript).

·         A figure that highlights the modes of actions of all listed drugs would be helpful. If only to demonstrate that most drugs affect several mitochondrially-relevant pathways/pathologies simultaneously. It would force the authors to consider causation. For example, oxidative stress leads to a fission phenotype as a protective measure associated with reduced ATP synthesis….

·         CoQ10 and curcumin are not mitochondrially targeted drugs. They go to them mitochondria but also everywhere else.

Author Response

Answer 1.

Answer: The authors would like to thank our reviewer for their helpful and constructive comments. They have truly guided us in reshaping and restructuring our manuscript. We arranged our text accordingly. We initially explain the pathophysiological mechanisms of mitochondrial dysfunction in kidney disease, then we discuss each drug in detail and bring a critique to most drugs in line with the available evidence and clinical trials. Finally, in our future perspectives section we present several additional trials which use interventions other then drugs. We’ve added a new table which explains the mechanisms of the drugs together with the available evidence derived from animal models and clinical trials.

Answer 2.

Thank you for your comment, we re-arranged our manuscript’s structure. We added each relevant clinical trial to the related drug and provided critical remarks to most available therapeutic option. In regard to your example (i.e. SS31), we added :

“SS-31 has shown promising results in several preclinical kidney disease models, however its transition into the clinics has been limited. One important barrier for the clinical use of SS-31 is its peptide structure. The difficulty to use peptides (i.e. SS-31) for chronic treatment could pose a significant limitation as peptides may not be directly suitable for use as convenient therapeutics as they potentially possess several intrinsic weaknesses such as poor chemical and physical stability, and a short circulating plasma half-life  (112). Nevertheless, these aspects need to be addressed for their use as clinical drugs (112). Further models are required to better understand the underlying mechanisms, signaling pathways and usefulness as well as to overcome the limitations of SS-31 in restoration of mitochondrial function and prevention of kidney disease. There are currently 25 clinical trials on SS-31 among which two have been studied on kidney disease patients (NCT01755858, NCT02436447). In a pilot phase 2a clinical trial (NCT01755858), co-administration of SS-31 with percutaneous transluminal renal angioplasty was found to attenuate post-procedural hypoxia, increase blood flow and kidney function (113). However, the results of the other study have not been reported. Given these findings SS-31 is a suitable molecule in kidney disease patients, however further pre-clinical and large-scale clinical studies are needed to overcome its certain drawbacks.”

In addition, we deleted the second SS-31 heading under “Anti-oxidants”.

Answer 3.

Thank you, as stated we revised our drugs section adding the related clincal trials as well as a critique to most of the available drugs. For curcumin and other antioxidants we added and edited the following: “Curcumin has been investigated in a large number of trials. However, although it had promising preclinical data, the clinical effectiveness of the native curcumin is weak (170). Similar to Curcumin, despite their efficiency and promising findings in experimental models of kidney dysfunction, the potential of mitochondria-targeted antioxidants as clinical treatments for kidney disease has not been promising and awaits further investigation. Several challenges remain for the translation of antioxidants into the clinics (6). In cell stress response pathways low and moderate levels of mitochondrial reactive oxygen species (mtROS) may function as signaling molecules (6, 171). Antioxidants can potentially have negative effects by scavenging and removing physiological levels of ROS involved in these signaling pathways (6). The measurement of ROS levels before or during the administration of mitochondria-targeted antioxidants might potentially be beneficial in preventing adverse effects and enable the safe use of these therapies (6). However, further developments are also required in these measurement methods and necessary equipment.”   Answer 4. Thank you for your detailed feedback. We have revised our manuscript accordingly. First we added the following under our Resveratrol heading: “There are currently six studies regarding the role of Resveratrol in kidney disease (NCT02433925, NCT03352895, NCT03597568, NCT03815786, NCT02704494, NCT03946176). One study (NCT02433925) investigated the effects of resveratrol on inflammation and oxidative stress in patients undergoing conservative treatment of CKD. Another study (NCT03352895) evaluated the effects of resveratrol on hearing impairment in patients with CKD on hemodialysis. However, among the given trials only, Sattarinezhad et al. (NCT02704494) (150) have reported their results which potentially indicates an absence of positive results.” Furthermore, we discussed the current clinical situation of MitoQ as: “There are currently 26 trials on MitoQ, among which two involve kidney disease patients (NCT02364648, NCT03960073). Earlier trials with MitoQ involving Parkinson’s Disease and Multiple Sclerosis did not report results, potentially suggesting, as in the case of Resveratrol, an absence of positive results. NCT03960073 is investigating the role of mitochondrial oxidative stress on exercise capacity and arterial hemodynamics in HFpEF patients with and without chronic kidney disease, this trial is still ongoing whereas the other study (NCT02364648) has not reported its results. Given these findings MitoQ has not shown significant promise regarding its role in the clinics. However, its preclinical results are somewhat promising and future studies are needed to better understand its effects in kidney disease.” We also discussed the situation of most the other drugs in the text.   Answer 5.

Thank you. We revised this section. As stated, we discussed the trials with their respective drugs and chemicals. Therefore we have indeed provided an in depth presentation of the current trials and have critiqued the current drugs and molecules accordingly.

In our future directions section we only list the trials having alternative interventions other than drugs therefore,  we added the following: “In addition to the clinical trials studying the effects of certain chemicals and drugs in kidney disease, there are several additional trials investigating mitochondrial dysfunction in CKD (NCT02923063), AKI (NCT05264285, NCT05458063), DKD (NCT04074668, NCT05319990, NCT05071287) and PKD (NCT04630613).” We will keep these trials as they investigate mitochondrial dysfunction in these kidney disease states, however do not use drugs or chemicals as the intervention point. These trials provide an additional perspective to mitochondrial dysfunction in kidney disease.

Answer 6.

We would like to thank our reviewer for their detailed critique and comment (s). They have truly helped us to re-shape and improve our manuscript. As stated we added critical comments to most drugs, added each relevant clinical trial to the related drug and discussed the available trials in more detail. Regarding this particular comment, we once again would like to express our gratitude and thank our reviewer.   Answer 7. Thank you. We revised this section. As stated, we discussed the trials with their respective drugs and chemicals. Here we added the following: “In addition to the clinical trials studying the effects of certain chemicals and drugs in kidney disease, there are several additional trials investigating mitochondrial dysfunction in CKD (NCT02923063), AKI (NCT05264285, NCT05458063), DKD (NCT04074668, NCT05319990, NCT05071287) and PKD (NCT04630613).” We will keep these trials as they investigate mitochondrial dysfunction in these kidney disease states, however do not use drugs or chemicals as the intervention point. These trials provide an additional perspective to mitochondrial dysfunction in kidney disease.   Answer 8.

Thank you, we deleted this trial, revised it as : “In addition to the clinical trials studying the effects of certain chemicals and drugs in kidney disease, there are several additional trials investigating mitochondrial dysfunction in CKD (NCT02923063), AKI (NCT05264285, NCT05458063), DKD (NCT04074668, NCT05319990, NCT05071287) and PKD (NCT04630613).”

  • A figure that highlights the modes of actions of all listed drugs would be helpful. If only to demonstrate that most drugs affect several mitochondrially-relevant pathways/pathologies simultaneously. It would force the authors to consider causation. For example, oxidative stress leads to a fission phenotype as a protective measure associated with reduced ATP synthesis….

Answer 9: Thank you, we added a new figure which explains the mechanisms of action of each drug.

Answer 10.

Thank you for bringing this up, although they may have broader actions, both also act on the mitochondria, therefor we will keep them where they are, under our anti-oxidants heading.